# Prevalence and risk factors of neurodevelopmental disorders among migrant and refugee preschool children in high-income western countries: A systematic review and meta-analysis protocol

Kh Shafiur Rahaman [1,2,3]*, Valsamma Eapen[4,5,6], Mythily Subramanium[7,8,9], James Rufus John[4,5,6], Kanchana Ekanayake[10], Amit Arora[1,2,11,12,13,14]

1 Discipline of Public Health, School of Medicine, Faculty of Health, Western Sydney University, Campbelltown Campus, Locked Bag, Penrith, New South Wales, Australia, 2 Health Equity across Lifespan Research Laboratory, Campbelltown, New South Wales, Australia, 3 Bangladesh Academy of Dietetics and Nutrition (BADN), Dhaka, Bangladesh, 4 School of Clinical Medicine, UNSW Australia, Liverpool, New South Wales, Australia, 5 Ingham Institute of Applied Medical Research, Liverpool, New South Wales, Australia, 6 South Western Sydney Local Health District, Liverpool, New South Wales, Australia, 7 Research Division, Institute of Mental Health, Singapore, Singapore, 8 Saw Swee Hock School of Public Health, National University of Singapore, Singapore, Singapore, 9 Lee Kong Chian School of Medicine, Nanyang Technological University, Singapore, 10 University of Sydney Library, Camperdown, New South Wales, Australia, 11 Translational Health Research Institute, Western Sydney University, Penrith, New South Wales 2751, Australia, 12 Discipline of Child and Adolescent Health, Sydney Medical School, Faculty of Medicine and Health, The University of Sydney, Westmead, New South Wales 2145, Australia, 13 Oral Health Services, Sydney Local Health District and Sydney Dental Hospital, NSW Health, Surry Hills, New South Wales 2010, Australia, 14 Aureolin Research, Consultancy, and Expertise Development (ARCED) Foundation, Dhaka, Bangladesh

* shafiur.mr@outlook.com

## Abstract

Neurodevelopment is a complicated mechanism involving genetic, cognitive, emotional, and behavioural processes. Factors related to parental migration directly or indirectly affect their children's neurodevelopmental process and may lead to neurodevelopmental disorders (NDDs). Other factors, such as barriers to accessing health services, social discrimination, mothers' psychosocial health during pregnancy may disrupt the neurodevelopmental process and lead to disorders and disabilities among children of migrants. However, there is a gap in data on the prevalence and the risk factors of neurodevelopmental disorders among migrant children, which have been inadequately listed. This paper presents a protocol for a systematic review to study and synthesise published evidence to ascertain the global prevalence of neurodevelopmental disorders and risk factors leading to those groups of neurodevelopmental disorders among children of migrants in high-income Western countries. The protocol for this systematic review was developed with guidance from the Joanna Briggs Institute (JBI) methodology and reported as per the Preferred Reporting Items for Systematic Review and Meta-Analysis Protocols (PRISMA-P) statement.

**Data availability statement:** No datasets were generated or analysed during the current study. All relevant data from this study will be made available upon study completion.

**Funding:** The research was funded by the "Western Sydney University International Master of Research (MRes) Scholarship". The funders had no role in study design, data collection and analysis, decision to publish, or preparation of the manuscript.

**Competing interests:** The authors have declared that no competing interests exist.

Observational studies that report on the prevalence and risk factors of neurodevelopmental disorders among migrant young children under 5 years of age in high-income Western countries will be included in this study. Five electronic databases will be searched comprehensively (MEDLINE, EMBASE, CINAHL, PsycINFO, and Scopus). Two reviewers will independently screen, select studies, assess the methodological quality, and extract all relevant data subsequently. The systematic review and meta-analysis will help design tailored interventions for migrant and refugee preschool children with neurodevelopmental disorders and identify gaps from previous research to guide future research. This review is registered with PROSPERO (CRD42024589357).

## 1. Introduction

Neurodevelopmental disorders (NDDs) involve a group of conditions that start in the development phase of a child and disrupt the individual, socioeconomic, academic, and occupational functioning [1]. Neurodevelopmental Disorders broadly include intellectual disorders, autistic spectrum disorders (ASD), attention deficit hyperactivity disorders (ADHD), language and learning disorders, motor disorders, etc. [1]. Both genetic and environmental factors contribute to the development of these disorders [2], leading to impairments in motor, cognitive, language, and behaviour [3]. Globally, 316.8 million cases of developmental conditions have been reported, with males being affected more than females [3]. The prevalence of neurodevelopmental disorders among children aged between 0–18 years differs across different population groups. A recent systematic review reported a significant variation in the prevalence of neurodevelopmental disorders (4.7% to 88.5%) [4]. The prevalence of Neurodevelopmental disorders in high-income countries was estimated between 7.0% to 15.0% of children in the general population [5,6]. In contrast, the prevalence in low-middle countries has been reported to be 7.6% [7], and children from low-economic settings are asymmetrically affected by neurodevelopmental disorders [8].

Various risk factors of neurodevelopmental disorders have been reported in the literature, which include genetic, environmental, prenatal, and postnatal issues, nutritional disorders, exposure to crisis, trauma, and poverty [8]. Furthermore, some children are more likely than others to develop developmental disabilities because of multiple unfavourable circumstances at different levels, which include but are not limited to the family's socioeconomic standing, their nationality, and their racial and ethnic heritage [3]. A recent review reported that children from migrant and refugee families coming from low and middle-income countries have a higher risk of neurodevelopmental disorders [9]. Children with one or both migrant parent(s) were found to have an increased risk of neurodevelopmental disorders related to language, academic skills, or coordination [10,11].

The risk factors related to neurodevelopmental delays in children may vary based on parents' ethnic background and circumstances during the migration [12,13]. Other difficulties related to stigma [14], inadequate knowledge, differences in perception [15], referral patterns [16], and delayed diagnosis [17] also contribute to the risk of

neurodevelopmental disorders among children with parental history of migration and ethnic minority. Migrants often exhibit risk factors such as inappropriate residence, inadequate nutrition (e.g., obesity) [18,19], and low socioeconomic status [20], which differ from those of the host population. Maternal mental health is another risk factor for neurodevelopmental disorders. Some migrant mothers also would have experienced war and political conflict or extreme economic crises in their home country. Some mothers may have been exposed to poverty and sexual violence, which caused high mental stress during their pregnancy [14]. Health literacy among migrant families varies, as do the developmental outcomes of children. Additionally, cultural sensitivity and the capacity of health services in the host country to identify and treat developmental disabilities and impairments can differ significantly.

Available data on the prevalence and incidence rates of neurodevelopmental disorders are necessary to plan for the sustainable delivery of health, social, and education services. However, data is often available only for the mainstream population, with only a few studies reporting on children with migrant and refugee backgrounds [12, 21–24]. The main objective of this study is to conduct a systematic review and/or meta-analysis of the literature to investigate the prevalence of neurodevelopmental disorders and their risk factors in preschool children (0–6 years) with a parental history of migration in high-income Western countries.

## 2. Methods

### 2.1. Protocol and registration

The systematic review will be conducted using the Joanna Briggs Institute (JBI) methodology for systematic reviews of effectiveness [25] and reported in accordance with the PRISMA (Preferred Reporting Items for Systematic Reviews and Meta-Analyses) guidelines [26]. The protocol for this review was reported as per the PRISMA for Systematic Review Protocols (PRISMA-P) framework. Additionally, the review has been registered with PROSPERO (CRD42024589357).

### 2.2. Review question

The primary review questions were designed based on the concept, context, and population (CoCoPop) framework as recommended by the Joanna Briggs Institute (JBI) [27]. The framework will suggest the search strategy and enable us to establish the inclusion and exclusion criteria (Table 1).

Among preschool children of migrant and refugee populations in high-income Western countries, what is the prevalence of neurodevelopmental disorders, and what are the risk factors affecting the disorders in question?

### 2.3. Inclusion criteria

"Migrants and refugees" refer to groups of people moving for numerous reasons, often with overlapping motivations, as part of mixed movements – as defined by the United Nations High Commissioner for Refugees (UNHCR) [30]. According to the International Organization for Migration (IOM), a "migrant" is anyone who crosses an international border, leaving their birthplace or usual place of residence, for instance, unauthorised entrants, asylum seekers, and irregular or undocumented migrants. This applies regardless of their legal status and includes authorised migrants who left for work, family,

**Table 1. Framework for inclusion criteria.**

| Parameter | | Criteria |
|---|---|---|
| **Co** | Concept | a. Prevalence of neurodevelopmental Disorders as per the DSM-5<br>b. Risk factors/ predictors of neurodevelopmental disorders |
| **Co** | Context/ setting | High-income western countries, as defined by the World Bank [28] |
| **Pop** | Population | Preschool children of migrants and refugees (as defined by the IOM [29], aged 0–6 years [mean/ median age of 6 years], with at least one parent born overseas (first or second generation of immigrants). |

and study [29]. Additionally, "refugee" encompasses individuals who have been granted refugee status or humanitarian protection and those fleeing persecution or organised violence – as defined by IOM [29]. In this study, children of migrants and refugee populations residing in high-income Western countries will be included, as classified by the World Bank (countries with a Gross National Income per capita of USD 13,205 or more) [28]. In this review, 'high-income Western countries' refers to nations in North America, Western Europe, and Oceania that are classified as high-income by the World Bank [28] and represent major destinations for migrant and refugee resettlement through different migration path-ways, such as skilled, family, and specific eligibility categories, along with refugee and humanitarian initiatives [31,32]. Individuals migrating from Africa, the Middle East, and Asia to Western nations often originate from collectivist societies that are transitioning towards individualist societies.

For our review, studies that included children aged between 0 and 6 years will be considered, or the studies that reported a mean/ median age of 6 years will be included, with at least one parent born overseas (first or second gener-ation of immigrants). Observational studies, such as cross-sectional, case-control, cohort studies (prospective and ret-rospective), and ecological studies, that report the prevalence of neurodevelopmental disorders and risk factors among children of migrants, will also be included. Other studies, such as randomised controlled trials, case reports, opinions, letters to editors, and unpublished materials, will be excluded. No restrictions will be applied to the publication date and the language of the published literature.

## 2.4. Search strategy

The search strategy will be developed with opinions from two expert health sciences librarians. A mix of specific medical subject headings (MeSH), free-text terms, and Boolean operators will be included. The search terms will focus on preva-lence, risk factors, migrants, refugees, preschool children, and relevant high-income Western countries. First, these terms will be tested in the Medline (OVID) database. The reviewers (K.S.R and A.A.) with experience in database searches will conduct an initial pilot search on two databases. The reviewers will then carry out all remaining literature searches inde-pendently. The preliminary Medline (OVID) search strategy formulated with guidance from expert librarians in the field is outlined in Table 2. After completing the Medline (OVID) database search, we will use the same syntax and subject head-ings for the remaining databases, adapting them as necessary. We will manually review the reference lists of the qualify-ing studies and existing literature reviews, including background and forward citation tracking of the studies included.

## 2.5. Information sources

The following five electronic databases will be searched without any restrictions on publication date (i.e., from the time of database inception to the present) and language: MEDLINE (Ovid), Embase (Ovid), PsycINFO (Ovid), Scopus, and Cumulative Index to Nursing and Allied Health Literature (CINAHL) (EBSCO).

## 2.6. Study selection

The studies identified through electronic databases and citations will be imported into EndNote 21 (Clarivate Analytics, Philadelphia, PA, USA). All the duplicates will be removed. Following a pilot test, two independent reviewers (K.S.R. and A.A.) will review the titles and abstracts, strictly adhering to the inclusion and exclusion criteria. The full text will be retrieved for clarification if necessary. Then, the articles that meet the inclusion criteria will be retrieved in full, and their details will be imported into the JBI System for Unified Management, Assessment and Review of Information (JBI SUMARI) [33]. The same two reviewers will then independently evaluate the full texts to confirm eligibility. When we need additional information, the study authors will be contacted. Any disagreements with the reviewer will be addressed through discussion involving a third reviewer (J.J.) if necessary. The reasons for excluding full-text studies will be recorded and reported in the systematic review, no formal hierarchy will be applied. If we find multiple articles published from the same

**Table 2. Primary search strategy for Medline (OVID) (4th October 2025).**

| # | Query | Results |
|---|---|---|
| 1 | Prevalence/ | 379136 |
| 2 | Risk Factors/ | 1052605 |
| 3 | (Prevalence* or risk factor*).mp. | 23479980 |
| 4 | 1 or 2 or 3 | 2347998 |
| 5 | exp Neurodevelopmental Disorders/ | 227526 |
| 6 | exp Communication Disorders/ | 72396 |
| 7 | exp Autism Spectrum Disorder/ | 49825 |
| 8 | exp "Attention Deficit and Disruptive Behavior Disorders"/ | 42946 |
| 9 | exp Specific Learning Disorder/ | 10679 |
| 10 | exp Motor Disorders/ | 1165 |
| 11 | exp Tic Disorders/ | 6332 |
| 12 | (Intellectual* Disabilit* or Global Developmental Delay* or Communication Disorder* or Language Disorder* or Speech Sound Disorder* or Phonological Disorder* or Childhood-Onset Fluency Disorder* or Stuttering* or Communication Disorder* or Autism* or autistic* or Attention-Deficit* or Hyperactivity Disorder* or Learning Disorder* or Motor Disorder* or Developmental Coordination Disorder* or Stereotypic Movement Disorder* or Tic Disorder* or Tourette* Disorder* or Motor disorder* or Tic Disorder* or Neurodevelopmental Disorder* or mp. | 250838 |
| 13 | 5 or 6 or 7 or 8 or 9 or 10 or 11 or 12 | 339043 |
| 14 | exp "Transients and Migrants"/ | 15835 |
| 15 | (migrant* or immigrant* or emigrant* or migration*).mp. | 455343 |
| 16 | "Culturally and linguistically".mp. | 2324 |
| 17 | Cultural diversity/ | 14119 |
| 18 | (cultural* divers* or ethnic*).mp. | 269245 |
| 19 | Ethnicity/ | 78681 |
| 20 | ethnic*.mp. or "Ethnic and Racial Minorities"/ | 255463 |
| 21 | Refugee*.mp. or Refugees/ | 21771 |
| 22 | 14 or 15 or 16 or 17 or 18 or 19 or 20 or 21 | 726013 |
| 23 | child/ or child, preschool/ or infant/ | 2574262 |
| 24 | (infant* or newborn* or child*).mp. | 3763723 |
| 25 | 23 or 24 | 3763723 |
| 26 | (high* income* western* adj5 (countr* or nation* or economy)).mp. | 101 |
| 27 | (western adj (countr* or econom* or nation*)).mp. | 22245 |
| 28 | (America* or Andorra* or Australia* or Austria* or Belgium or Belgian* or Britain or British or Canada or Canadian* or Croatia* or Cyprus or Cyprian* or Cypriot? or Denmark or Danish or England or English or Estonia* or Finland or Finnish or Finn? or France or French or German* or Greece or Greek* or Hungary or Hungarian*or Iceland* or Ireland or Irish or Italy or Italian* or Latvia* or Liechtenstein*or Lithuania* or Luxembourg* or Malta or Maltese or Monaco or Netherlands or Dutch or New Zealand* or Norway or Norwegian*or Poland or Polish or Portug* or San Marino or Slovak* or Slovenia* or Scotland or Scottish or Spain or Spanish or Sweden or Swedish or Switzerland or Swiss or United Kingdom or UK or Great Britain or United States or USA or Wales or Welsh).tw,kw. | 2969759 |
| 29 | (Americas or Andorra or Australia or Austria or Belgium or Canada or Croatia or Cyprus or Denmark or England or Estonia or Finland or France or Germany or Greece or ancient Greece or Hungary or Austria-Hungary or Iceland or Ireland or northern Ireland or Italy or Latvia or Liechtenstein or Lithuania or Luxembourg or Malta or Monaco or Netherlands or New Zealand or Norway or Poland or Portugal or San Marino or Slovakia or Czechoslovakia or Czech republic or Scotland or Spain or Sweden or Switzerland or United Kingdom or United States or Wales).sh. | 2624787 |
| 30 | 26 or 27 or 28 or 29 | 4446097 |
| 31 | 4 and 13 and 22 and 25 and 30 | 543 |

study, they will be linked. The study selection process will be presented as a Preferred Reporting Items for Systematic Reviews and Meta-Analysis (PRISMA) flow diagrams [34].

## 2.7. Assessment of methodological quality

Each study included in this review will be assessed distinctly by two reviewers (K.S.R and A.A.) for methodological quality. A standardised critical appraisal tool developed by the Joanna Briggs Institute (JBI) will be used to assess the methodological quality of the included studies [35], selected according to study design. The JBI Critical Appraisal Checklist for Studies reporting Prevalence Data will be used for studies reporting prevalence estimates of neurodevelopmental disorders [36]. Additionally, the JBI Critical Appraisal Checklist for cross-sectional studies, cohort studies and case-control studies will be used for assessing the risk or protective factors of neurodevelopmental disorders [37]. Any disagreements will be resolved through discussion between the two reviewers, and a third reviewer (J.J.) will be included when necessary. The authors of the study will be contacted to request missing or additional information where necessary to ensure the methodological quality. The studies will be assessed based on the available information if there is no response from the authors following two attempts. The risk of bias will be presented for each study through tables and figures and described narratively in the final review publication. The methodological concerns and their impact on the interpretation of results will be addressed in this narrative. All articles considered in this review will be subject to data extraction and synthesis, regardless of their methodological quality outcomes. Where possible, each study will be included for data extraction and synthesis irrespective of its methodological quality.

## 2.8. Data extraction

A standard data extraction tool (see Table 1 in the Appendix) has been developed, which will be piloted in one study. The tool will be refined to ensure that all relevant data is captured. A calibration exercise will be conducted to ensure consistency among the two reviewers. Two authors (K.S.R and A.A.) will independently extract the data. Any differences among reviewers will be settled through consensus, and if an agreement cannot be achieved, a third reviewer will make the final decision. The extracted data will be recorded in an Excel spreadsheet with article details, participant characteristics, prevalence and risk factors of neurodevelopmental disorders, type of migrant or refugee, study context, etc. Any additional information relevant to our study will be recorded, and the data extraction form will be modified accordingly. The authors of the study will be contacted when we require any additional information. If there is no response from the authors following two attempts, we will continue with the available information. The extracted data will be presented in narrative form and tables.

## 2.9. Data synthesis

Quantitative data synthesis methods will be employed to comprehensively address the research findings. The results will be categorised into settings, considering the origin of the participants and other environmental features. A meta-analysis will be performed only when there are at least two eligible studies available, and the studies included are relatively consistent regarding the population, outcomes, and methods used. For meta-analysis of prevalence estimates, we will use a random-effects model with the Freeman–Tukey double arcsine transformation to stabilize variances and reduce bias when proportions are close to 0 or 1 [38]. This transformation enables the inclusion of studies reporting zero events without the need for continuity corrections. Pooled prevalence will be presented with 95% confidence intervals. Heterogeneity will be assessed using the $I^2$ statistic and Cochran's Q test and explored through subgroup analyses and meta-regression where sufficient studies are available. For meta-analysis of risk factors, the preferred effect measure will be adjusted odds ratios (aORs) with 95% confidence intervals. Where studies report risk ratios (RRs) or hazard ratios (HRs), we will pool effect sizes within the same metric and study design to avoid inappropriate mixing. If conversion between effect measures is

required, we will apply standard statistical methods (e.g., log transformation and variance approximation) and document these procedures in detail. Random-effects models will be used for pooling, and heterogeneity will be assessed using the I² statistic and Cochran's Q test. Otherwise, we will report the findings narratively. We will conduct a subgroup analysis to explore sources of heterogeneity where sufficient data are available. A priori subgroups would include duration of resettlement (recently settled, < 5 years vs long-term, ≥ 5 years), geographical origin of the migrants & refugees (e.g., Middle East, South Asia, Africa, other), region of the host country (e.g., North America, Europe, Oceania etc.) to address contextual variability, age group (infants, < 2 years vs preschool, 2–6 years), and type of neurodevelopmental disorders such as autism, learning disorders, ADHD and others.

Subgroup analyses will only be conducted when there are ≥ 5 studies per subgroup, and meta-regression will require ≥10 studies to ensure adequate power and reliability. These subgroups could be more vulnerable to health conditions due to specific socioeconomic conditions, access to healthcare, and health inequalities. To evaluate publication bias, funnel plots will be utilized for meta-analyses that include 10 or more studies. Egger's regression test will be conducted only when there are at least 10 studies, in line with recommendations to prevent false positives with smaller sample sizes [39]. Studies identified with a high risk of bias will still be included in the review, but with careful interpretation. These studies will not be part of the primary meta-analyses and will instead be evaluated in sensitivity analyses to determine their influence on the pooled estimates. Should the inclusion of these studies significantly change the summary estimates or the level of heterogeneity, the findings will be presented separately and discussed in context. A narrative synthesis of the study findings will be used in tables and figures where statistical pooling is challenging due to heterogeneity among the studies. Stata 18 will be used for any statistical analysis to ensure rigorous data evaluation, leading to more valid and reliable conclusions.

## 3. Conclusions

The systematic review aims to provide overarching details on the prevalence of neurodevelopmental disorders and risk factors among preschool children of migrants (aged 0–6 years) in a global context. Understanding the risk factors is crucial to knowing what contributes to this population's higher prevalence of neurodevelopmental disorders. The findings can help to design targeted interventions, develop culturally appropriate programs, and attenuate these risks. Policymakers and health officials can use the review findings to address the unique challenges faced by children and their migrant parents within the health system. Our research also prioritises the necessity of including a diverse group in health research. The migrant population has its unique cultural practices, norms, and traditions. In this review, we aim to address the insufficiently represented group and contribute to equitable health research practices.

## Supporting information

**S1 File. Supporting information.**
(DOCX)

## Acknowledgments

This study acknowledges the support from the School of Health Sciences (SoHS) and Graduate Research School (GRS), Western Sydney University.

## Author contributions

**Conceptualization:** Kh Shafiur Rahaman, Valsamma Eapen, James Rufus John, Amit Arora.

**Methodology:** Kh Shafiur Rahaman, James Rufus John, Amit Arora.

**Supervision:** Mythily Subramanium, Amit Arora.

**Writing – original draft:** Kh Shafiur Rahaman.

**Writing – review & editing:** Kh Shafiur Rahaman, Valsamma Eapen, Mythily Subramanium, James Rufus John, Kanchana Ekanayake, Amit Arora.

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
