## [Editor Report · Decision Letter 0]

13 Dec 2024

PONE-D-24-50733

Prevalence and risk factors of neurodevelopmental disorders among migrant and refugee preschool children in high-income countries: A systematic review and meta-analysis protocol

PLOS ONE

Dear Dr. Kh Shafiur Rahaman,

Thank you for submitting your manuscript to PLOS ONE. After careful consideration, we have decided that your manuscript does not meet our criteria for publication and must therefore be rejected.

Specifically:

The study addresses a very interesting topic and the systematic review has also been accepted on PROSPERO. However, the results, the forest plot of the meta-analysis, the discussion are missing.

I am sorry that we cannot be more positive on this occasion, but hope that you appreciate the reasons for this decision.

Kind regards,

Massimiliano Esposito, M.D.

Academic Editor

PLOS ONE

---

## [Author Response · Author response to Decision Letter 1]

27 Dec 2024

I have updated the cover letter and wish to consider this manuscript a Study Protocol. I am also confirming that the literature searches have not been completed.

---

## [Decision Letter · Decision Letter 1]

12 Sep 2025

Please find below my overall assessment and detailed comments on the current version.

The protocol addresses an important evidence gap and has strong potential to inform policy and service planning for migrant and refugee children. The manuscript is clearly written and this revision clearly improved the manuscript, but several key issues remain unresolved. Some of these were highlighted by the first reviewer and still appear unaddressed, while others are additional concerns identified during my own review. Collectively, these points require clarification and revision before the manuscript can be considered for publication.

**Comments:**

Age range is inconsistently described as “under 5,” “<6,” and “preschool.” Please standardize across the title, abstract, methods, and eligibility table.The phrase “high-income western countries” is ambiguous. If all high-income countries (per World Bank) are eligible, state this clearly. If only a subset of “western” countries are intended, define them explicitly and ensure the search strategy matches.Clarify whether you will include first-generation children only (born abroad) or also second-generation children (born locally to migrant/refugee parents).Different sections provide contradictory statements: “English only” vs “no restrictions.” Please decide on one policy. If restricted to English, provide a rationale and acknowledge risk of language bias.The Medline strategy includes many non-NDD conditions (e.g., schizophrenia, depression, anxiety, bipolar disorder, substance use, sleep disorders). This will generate excessive irrelevant records and reduce precision. Please restrict to DSM-5 neurodevelopmental disorders: autism spectrum disorder, ADHD, communication disorders, specific learning disorder, motor disorders, tic disorders, intellectual disability.The country filter includes outdated or irrelevant terms (e.g., “Czechoslovakia,” “ancient Greece,” “Austria-Hungary”). Please update to a current and justified list, or omit country filters and apply eligibility during screening.Database listing is inconsistent (sometimes four, sometimes five). Correct to a consistent list (MEDLINE, EMBASE, CINAHL, PsycINFO, Scopus).Please specify the exact JBI tools by study type (e.g., JBI Prevalence checklist for prevalence, JBI cohort/case-control/cross-sectional checklists for risk factors). Current phrasing “relevant JBI tool” is too vague.Clarify how studies at high/critical risk of bias will be handled in synthesis (e.g., excluded from quantitative pooling, included in sensitivity analysis only).Prevalence pooling: Specify the statistical transformation (e.g., logit, Freeman–Tukey, binomial GLMM) and how zero-event studies will be handled.Risk factor pooling: State the preferred effect measure (e.g., adjusted odds ratios) and how you will handle mixing ORs, RRs, and HRs.Subgroups and meta-regressions: Define a priori variables more clearly (e.g., “recently resettled <5 years” vs “long-term”), and state minimum study counts required to conduct analyses.Publication bias: Egger’s test should only be used with ≥10 studies; please clarify this.The manuscript states support from a “Western Sydney University International MRes Scholarship,” while the submission form says “no funding.” Please align these.Data availability should not be “Not applicable.” A protocol-appropriate statement would be: “No datasets were generated or analyzed. Data extraction forms and analytic code will be shared upon completion of the review.”

Other issues:

Methods: replace “Total text will be retrieved” with “Full text will be retrieved.”Data extraction table: “Reviewer comments” column is unnecessary for publication; simplify to standard fields.Clarify in Methods that two reviewers will independently extract data and resolve discrepancies by consensus.References: check for duplicates and fix DOI formatting errors.Standardize terminology: use either “neurodevelopmental disorders (NDDs)” or “neurodevelopment disorders” consistently.

We look forward to receiving your revised manuscript.

Kind regards,

Selcuk Guven, Ph.D.

Guest Editor

PLOS ONE

Journal Requirements:

"Western Sydney University International Master of Research (MRes) Scholarship."

5. We note you have included a table to which you do not refer in the text of your manuscript. Please ensure that you refer to Table 1 in your text; if accepted, production will need this reference to link the reader to the Table.

6. We notice that your supplementary tables are included in the manuscript file. Please remove them and upload them with the file type 'Supporting Information'. Please ensure that each Supporting Information file has a legend listed in the manuscript after the references list.

Additional Editor Comments (if provided):

Reviewer #1:

Reviewers' comments:

Reviewer's Responses to Questions

**Comments to the Author**

1. Does the manuscript provide a valid rationale for the proposed study, with clearly identified and justified research questions?

Reviewer #1: Yes

2. Is the protocol technically sound and planned in a manner that will lead to a meaningful outcome and allow testing the stated hypotheses?

Reviewer #1: Yes

3. Is the methodology feasible and described in sufficient detail to allow the work to be replicable?

Reviewer #1: Yes

4. Have the authors described where all data underlying the findings will be made available when the study is complete?

Reviewer #1: Yes

5. Is the manuscript presented in an intelligible fashion and written in standard English?

Reviewer #1: Yes

You may also provide optional suggestions and comments to authors that they might find helpful in planning their study.

Reviewer #1: Thank you for inviting me to review this manuscript. This article is well written and the protocol is robust and thorough due to the guidance of the JBI, PRISMA-P, and PRISMA guidelines. The analytical plan is also very robust with important subgroup and sensitivity analyses, and the assessment of publication bias. I have made some comments below for the authors to consider and to hopefully strengthen the manuscript.

Comments/suggestions:

• General:

o No need to capitalise “neurodevelopment/al” in the middle of sentences

o It should be “neurodevelopmental disorder”, not “neurodevelopment disorder”

o There are inconsistencies in how the target population is mentioned (e.g., “<6 years”, “children between 0 and 5 years”, “children under 5 years of age” and “0 to 6 years”). For consistency, clarity, and ease of reading, perhaps stick to one.

• Abstract:

o Line 52: The second last sentence seems to be incomplete (i.e., “migrant children with and”)

• Introduction:

o Lines 69-71: The third sentence seems to be incomplete (i.e., “Both genetic and environmental factors contribute to the development of these disorders [2], leading to impairments in motor, cognitive, language, and behavioural [3]”)

o Lines 100-116: Remove specific research questions or consolidate with Methods section

• Methods:

o Table 1. Framework for inclusion criteria:

I understand that high-income countries were defined as per the World Bank. How have you selected the “Western countries” as this is slightly ambiguous? Were they selected based on geography (i.e., continent, region) and/or cultural, political, or economic characteristics? Do you have a reference?

o 2.3 Inclusion Criteria:

Lines 143-151: Perhaps for succinctness in this section, the description of migrants and refugees should be moved to the Introduction section.

Lines 155-157: What about other study designs such as case-control, time-series, or ecological studies?

o 2.5 Information Sources:

Line 184: I believe PsycINFO is also a part of the Ovid platform

o 2.6 Study Selection:

Lines 196-197: Is there any hierarchy for reasons for exclusion for the full-text screening stage?

o 2.7 Assessment of Methodological Quality:

Line 204: It should be “risk of bias”, “calibre” is not a suitable term as it is not informative and non-specific

o 2.9 Data Synthesis:

Before mentioning the weighted inverse variance random-effects model, I would mention that you will perform the meta-analysis if the number of eligible studies and homogeneity permits it.

How will heterogeneity be tested?

• Appendix:

o Table 1: Data extraction template.

Maybe be worth including the study period in this table.

**Do you want your identity to be public for this peer review?** For information about this choice, including consent withdrawal, please see our Privacy Policy

Reviewer #1: No

---

## [Author Response · Author response to Decision Letter 2]

13 Oct 2025

Response to Editor’s Comments

The protocol addresses an important evidence gap and has strong potential to inform policy and service planning for migrant and refugee children. The manuscript is clearly written and this revision clearly improved the manuscript, but several key issues remain unresolved. Some of these were highlighted by the first reviewer and still appear unaddressed, while others are additional concerns identified during my own review. Collectively, these points require clarification and revision before the manuscript can be considered for publication.

Response:

We sincerely thank the Academic Editor for the careful evaluation of our revised manuscript and for recognizing the significance and potential policy implications of our protocol. We are grateful for the acknowledgment that the manuscript addresses a vital evidence gap and that the revisions have strengthened the clarity and quality of the paper.

We appreciate the editor’s observation that several key issues remain unresolved. We have carefully reviewed each point raised in both the current and previous reviews and have comprehensively addressed all outstanding concerns in this version of the manuscript.

In the subsequent paragraphs, we responded to each comment/query raised by the editor and reviewer.

Comment 1. Age range is inconsistently described as “under 5,” “<6,” and “preschool.” Please standardize across the title, abstract, methods, and eligibility table.

Response: We thank the reviewer for identifying this critical point regarding consistency in the age range terminology. We have now standardized the description of the target population across the entire manuscript. The term “preschool children (0-6 years)” is used consistently across the manuscript. In the inclusion criteria, we have also specified a “mean/median age of 6 years” as the inclusion criteria, as many included studies provided this information.

Comment 2. The phrase “high-income western countries” is ambiguous. If all high-income countries (per World Bank) are eligible, state this clearly. If only a subset of “western” countries are intended, define them explicitly and ensure the search strategy matches.

Response: We appreciate the reviewer’s thoughtful comment. We acknowledge that the term “high-income western countries” could appear ambiguous; however, its use in our protocol is intentional and contextually valid. Our study focuses on migrant and refugee populations who predominantly resettle in high-income Western nations, such as those in North America, Western Europe, and Oceania, which collectively host the majority of global refugee and migrant populations from low- and middle-income countries.

This pattern of migration has been well documented in the literature. For instance, the United Nations High Commissioner for Refugees (UNHCR, 2023) and the OECD (2022) report that high-income Western countries, including the United States, Canada, the United Kingdom, Germany, France, and Australia, continue to be the primary destinations for resettlement and immigration. Moreover, the list of Western countries was obtained from an experienced librarian who was involved in the search.

Therefore, retaining the term “high-income Western countries” accurately reflects both the geographic and socioeconomic contexts most relevant to our research question. Many published studies have used the term “high-income Western countries”. Below are some of the published papers for your record:

1) https://www.sciencedirect.com/science/article/pii/S027795362400162X

2) https://journals.plos.org/plosone/article?id=10.1371/journal.pone.0170700

To ensure transparency, we have now clarified this in the revised manuscript, explicitly mentioning why we have chosen “Western countries” and citing relevant literature.

We have added “In this review, ‘high-income Western countries’ refers to nations in North America, Western Europe, and Oceania that are classified as high-income by the World Bank [31] and represent major destinations for migrant and refugee resettlement through different migration pathways, such as skilled, family, and specific eligibility categories, along with refugee and humanitarian initiatives. Individuals migrating from Africa, the Middle East, and Asia to Western nations often originate from collectivist societies that are transitioning towards individualist societies”.

Comment 3. Clarify whether you will include first-generation children only (born abroad) or also second-generation children (born locally to migrant/refugee parents).

Response: We appreciate the reviewer's insightful comment. We agree that clarifying the generational scope of inclusion is essential to ensure transparency and consistency. In the revised manuscript, we now explicitly state that both first-generation (foreign-born) and second-generation (locally born to migrant or refugee parents) children will be included.

Comment 4. Different sections provide contradictory statements: “English only” vs “no restrictions.” Please decide on one policy. If restricted to English, provide a rationale and acknowledge risk of language bias.

Response: We thank the reviewer for bringing this inconsistency to our attention. We have now standardized the language inclusion policy throughout the manuscript. The revised protocol specifies that no restrictions will be applied to language during the search.

Comment 5. The Medline strategy includes many non-NDD conditions (e.g., schizophrenia, depression, anxiety, bipolar disorder, substance use, sleep disorders). This will generate excessive irrelevant records and reduce precision. Please restrict to DSM-5 neurodevelopmental disorders: autism spectrum disorder, ADHD, communication disorders, specific learning disorder, motor disorders, tic disorders, intellectual disability.

Response: Thank you for this critical observation. Initially, we had included all the conditions under DSM-5; however, during the screening process, we considered neurodevelopmental disorders only. We agree that including non-NDD conditions (e.g., schizophrenia, depression, anxiety, bipolar disorder, substance use, sleep disorders) in the MEDLINE strategy retrieved irrelevant records. Therefore, we have revised the search strategy to restrict the condition block strictly to DSM-5 neurodevelopmental disorders, as recommended.

Comment 6. The country filter includes outdated or irrelevant terms (e.g., “Czechoslovakia,” “ancient Greece,” “Austria-Hungary”). Please update to a current and justified list, or omit country filters and apply eligibility during screening.

Response: Thank you for raising this point. We revisited Medline to review those country terms and found that MEDLINE still retains them as MeSH terms due to legacy and historical geographic descriptors. Keeping those terms can support retrieval of records indexed during geopolitical transition periods (e.g., early 1990S), also help missing relevant studies that were catalogued under old labels, as we did not have any restrictions to the date of publication.

Comment 6. Database listing is inconsistent (sometimes four, sometimes five). Correct to a consistent list (MEDLINE, EMBASE, CINAHL, PsycINFO, Scopus).

Response: We have carefully reviewed all sections of the manuscript and have now standardized the list of databases to ensure consistency throughout. The final and correct list of databases to be searched includes five databases: MEDLINE, EMBASE, CINAHL, PsycINFO, and Scopus.

Comment 7. Please specify the exact JBI tools by study type (e.g., JBI Prevalence checklist for prevalence, JBI cohort/case-control/cross-sectional checklists for risk factors). Current phrasing “relevant JBI tool” is too vague.

Response: We appreciate the reviewer's valuable suggestion. We agree that specifying the exact Joanna Briggs Institute (JBI) critical appraisal tools enhances the methodological transparency of the protocol. We have now clearly outlined the tools to be used in accordance with the study design. Specifically, the following JBI checklists will be applied:

1. JBI Critical Appraisal Checklist for Studies Reporting Prevalence Data – for studies reporting prevalence estimates of neurodevelopmental disorders.

2. JBI Critical Appraisal Checklist for Analytical Cross-sectional Studies – for studies examining associations between risk factors and neurodevelopmental disorders.

3. JBI Critical Appraisal Checklist for Case–control Studies – for studies comparing migrant/refugee children with control groups.

4. JBI Critical Appraisal Checklist for Cohort Studies – for longitudinal designs assessing risk or protective factors over time.

This level of specification ensures that each included study is appraised using the most appropriate tool for its design, in alignment with JBI methodology (Aromataris & Munn, 2020). We have revised the “quality assessment” section of the manuscript and provided relevant citations.

Comment 8. Clarify how studies at high/critical risk of bias will be handled in synthesis (e.g., excluded from quantitative pooling, included in sensitivity analysis only).

Response: Thank you for this helpful suggestion. We have clarified in the revised manuscript how studies with high or critical risk of bias will be managed. Specifically, such studies will not be excluded outright but will be included in sensitivity analyses to assess the robustness of pooled estimates. Their influence on the overall findings will be carefully interpreted with caution and discussed in the results and discussion sections.

Comment 9. Prevalence pooling: Specify the statistical transformation (e.g., logit, Freeman–Tukey, binomial GLMM) and how zero-event studies will be handled.

Response: Thank you for this important suggestion. We agree that specifying the statistical transformation and handling of zero-event studies will improve clarity and reproducibility. We have revised the Methods section to include these details. Specifically, we will use the Freeman–Tukey double arcsine transformation for pooling prevalence estimates, as it stabilizes variances when proportions are near 0 or 1. Zero-event studies will be retained in the analysis using this transformation, which accommodates such cases without continuity corrections. This approach is widely recommended for meta-analyses of proportions.

Comment 10. Risk factor pooling: State the preferred effect measure (e.g., adjusted odds ratios) and how you will handle mixing ORs, RRs, and HRs.

Response: Thank you for highlighting this point. We have revised the “data synthesis” section to state that adjusted odds ratios (aORs) will be the preferred effect measure for risk factor pooling, as they account for confounding. When studies report risk ratios (RRs) or hazard ratios (HRs), we will not mix them in the same meta-analysis. Instead, we will pool effect sizes within the same metric and study design. If conversion is necessary for synthesis, we will use established statistical methods (e.g., log transformation and variance approximation) and report these transparently.

Comment 11. Subgroups and meta-regressions: Define a priori variables more clearly (e.g., “recently resettled <5 years” vs “long-term”), and state minimum study counts required to conduct analyses.

Response: We agree that defining subgroup and meta-regression variables a priori and specifying minimum study counts will improve transparency and rigor. We have revised the relevant section to clearly list the planned subgroup variables and meta-regression covariates, along with the threshold for conducting these analyses. Specifically, subgroup analyses will be performed only when there are ≥5 studies per subgroup, and meta-regression will require ≥10 studies to ensure sufficient statistical power.

We have stated “We will conduct a subgroup analysis to explore sources of heterogeneity where sufficient data are available. A priori subgroups would include duration of resettlement (recently settled, <5 years vs long-term, ≥5 years), geographical origin of the migrants & refugees (e.g., Middle East, South Asia, Africa, other), region of the host country (e.g., North America, Europe, Oceania etc.), age group (infants, <2 years vs preschool, 2-6 years), and type of neurodevelopmental disorders such as autism, learning disorders, ADHD and others. Subgroup analyses will only be conducted when there are ≥5 studies per subgroup, and meta-regression will require ≥10 studies to ensure adequate power and reliability”.

Comment 12. Publication bias: Egger’s test should only be used with ≥10 studies; please clarify this.

Response: We have clarified in the Methods section that Egger’s regression test will only be conducted when there are ≥10 studies in the meta-analysis.

Comment 13. The manuscript states support from a “Western Sydney University International MRes Scholarship,” while the submission form says “no funding.” Please align these.

Response: We have corrected the discrepancy between the manuscript and the submission form. The manuscript now clearly states that the study is supported by the Western Sydney University International MRes Scholarship, and the submission form has been updated to reflect this funding source. This ensures alignment across all documents.

Comment 14. Data availability should not be “Not applicable.” A protocol-appropriate statement would be: “No datasets were generated or analyzed. Data extraction forms and analytic code will be shared upon completion of the review.”

Response: We have revised the Data Availability statement to reflect that no datasets were generated or analyzed at this stage, and that data extraction forms and analytic code will be shared upon completion of the review.

Other issues:

1. Methods: replace “Total text will be retrieved” with “Full text will be retrieved.”

Response: We have corrected the phrase in the Methods section to use “Full text will be retrieved,” which is the appropriate terminology for systematic reviews.

2. Data extraction table: “Reviewer comments” column is unnecessary for publication; simplify to standard fields.

Response: We have simplified the data extraction table to include only standard fields

3. Clarify in Methods that two reviewers will independently extract data and resolve discrepancies by consensus.

Response: We have updated the Methods section to explicitly state that two reviewers will independently extract data and resolve disagreements by consensus, with a third reviewer available if needed.

4. References: check for duplicates and fix DOI formatting errors.

Response: We have reviewed the reference list thoroughly, removed any duplicate entries, and corrected all DOI formatting errors to comply with journal standards

5. Standardize terminology: use either “neurodevelopmental disorders (NDDs)” or “neurodevelopment disorders” consistently.

Response: Thank you for pointing this out. We agree that consistent terminology is essential for clarity and accuracy. We have standardized the term throughout the manuscript to “neurodevelopmental disorders”.

Response to Reviewer

Reviewer #1: Thank you for inviting me to review this manuscript. This article is well written and the protocol is robust and thorough due to the guidance of the JBI, PRISMA-P, and PRISMA guidelines. The analytical plan is also very robust with important subgroup and sensitivity analyses, and the assessment of publication bias. I have made some comments below for the authors to consider and to hopefully strengthen the manuscript.

Response: Thank you for your positive feedback and constructive suggestions. We appreciate your acknowledgment of the robustness of the protocol and analytical plan. Below are our detailed responses to your comments:

Comments/suggestions:

General:

o No need to capitalise “neurodevelopment/al” in the middle of sentences

Response: We have corrected all instances where “Neurodevelopmental” was unnecessarily capitalized in the middle of sentences.

o It should be “neurodevelopmental disorder”, not “neurodevelopment disorder”

Response: We have standardized terminology throughout the manuscript to “neurodevelopmental disorders (NDDs)” in line with DSM-5.

o There are inconsistencies in how the target population is mentioned (e.g., “<6 years”, “children between 0 and 5 years”, “children unde

---

## [Editor Report · Decision Letter 2]

10 Nov 2025

Prevalence and risk factors of neurodevelopmental disorders among migrant and refugee preschool children in high-income western countries: A systematic review and meta-analysis protocol

PONE-D-24-50733R2

Dear Dr. Rahaman,

We’re pleased to inform you that your manuscript has been judged scientifically suitable for publication and will be formally accepted for publication once it meets all outstanding technical requirements.

Kind regards,

Selcuk Guven, Ph.D.

Guest Editor

PLOS ONE

Additional Editor Comments (optional):

Thank you for your careful and thorough revisions. You have successfully addressed all concerns raised in the previous review round, and the manuscript has been significantly strengthened as a result. I am pleased to inform you that the revised version now meets the journal’s standards, and I will recommend it for acceptance.
---

## [Editor Report · Acceptance letter]

PONE-D-24-50733R2

PLOS ONE

Dear Dr. Rahaman,

I'm pleased to inform you that your manuscript has been deemed suitable for publication in PLOS ONE. Congratulations! Your manuscript is now being handed over to our production team.

Kind regards,

on behalf of

Dr. Selcuk Guven

Guest Editor

PLOS ONE